# Downregulation of Ribosomal Protein Genes Is Revealed in a Model of Rat Hippocampal Neuronal Culture Activation with GABA(A)R/GlyRa2 Antagonist Picrotoxin

**DOI:** 10.3390/cells13050383

**Published:** 2024-02-23

**Authors:** Alexander Beletskiy, Anastasia Zolotar, Polina Fortygina, Ekaterina Chesnokova, Leonid Uroshlev, Pavel Balaban, Peter Kolosov

**Affiliations:** 1Institute of Higher Nervous Activity and Neurophysiology, The Russian Academy of Sciences, 117485 Moscow, Russia; apbeletskiy@mail.ru (A.B.); zolotar.ananas@gmail.com (A.Z.); fortyginapolina@mail.ru (P.F.); katyachesn@gmail.com (E.C.); leoniduroshlev@gmail.com (L.U.); pmbalaban@gmail.com (P.B.); 2Engelhardt Institute of Molecular Biology, The Russian Academy of Sciences, 119991 Moscow, Russia

**Keywords:** long-read sequencing, ONT MinION, RNA-Seq, primary neuronal cultures, neuronal plasticity, picrotoxin, ribosomal proteins, transposable elements

## Abstract

Long-read transcriptome sequencing provides us with a convenient tool for the thorough study of biological processes such as neuronal plasticity. Here, we aimed to perform transcriptional profiling of rat hippocampal primary neuron cultures after stimulation with picrotoxin (PTX) to further understand molecular mechanisms of neuronal activation. To overcome the limitations of short-read RNA-Seq approaches, we performed an Oxford Nanopore Technologies MinION-based long-read sequencing and transcriptome assembly of rat primary hippocampal culture mRNA at three time points after the PTX activation. We used a specific approach to exclude uncapped mRNAs during sample preparation. Overall, we found 23,652 novel transcripts in comparison to reference annotations, out of which ~6000 were entirely novel and mostly transposon-derived loci. Analysis of differentially expressed genes (DEG) showed that 3046 genes were differentially expressed, of which 2037 were upregulated and 1009 were downregulated at 30 min after the PTX application, with only 446 and 13 genes differentially expressed at 1 h and 5 h time points, respectively. Most notably, multiple genes encoding ribosomal proteins, with a high basal expression level, were downregulated after 30 min incubation with PTX; we suggest that this indicates redistribution of transcriptional resources towards activity-induced genes. Novel loci and isoforms observed in this study may help us further understand the functional mRNA repertoire in neuronal plasticity processes. Together with other NGS techniques, differential gene expression analysis of sequencing data obtained using MinION platform might provide a simple method to optimize further study of neuronal plasticity.

## 1. Introduction

It is known that about two-thirds of all genes are expressed in mammalian neuronal tissue [1]. Neuronal cells are also very elongated and highly compartmentalized; axon, dendrites, and soma may contain different RNA sets. In these cells, transcription intensity and translational status of a given mRNA or an mRNA pool may be different depending on neuronal activation, signals from other neurons, extracellular conditions, and many other factors [2,3]. Taken together, these facts suggest that precise spatial and temporal regulation of RNA synthesis is necessary for normal neuronal function. Importantly, some mechanisms of transcription specific to neurons may be easily overlooked when the most common RNA-Seq methods are used.

In addition, the repertoire of mRNAs presented in neurons is further enriched by alternative splicing characteristics for the processes of neuronal plasticity. Neuronal activation may serve as a trigger for the final mRNA maturation step, so the mRNA formation is completed right before the start of the protein translation. For example, the longest isoform (Calm3_L_) of calmodulin 3 mRNA is localized to neuronal dendrites, while lack of the 3′-UTR retained intron impairs its targeting to dendrites. NMDA-mediated synaptic activation specifically promotes the distal dendritic mRNA localization of the Calm3_L_ isoform [4]. Another example is Arc mRNA. Its abundance is associated with synaptic plasticity and regulated by the presence or absence of two 3′-UTR introns [5]. Notably, neurons and neuronal precursors have increased retrotransposon mobilization rates and expression of retrotransposon mRNAs compared with other cell types; this contributes to neuronal somatic mosaicism [6,7,8] and has a role in neuronal differentiation [9].

Another peculiarity about transcription in neurons is that transcription and translation processes may be uncoupled, because there are multiple mechanisms of mRNA conservation and its relocation to dendrites where translational machinery is relatively autonomous [10]. It was also shown that damaged dendrite ribosomes may be promptly restored on site by replacing individual ribosomal proteins using local translation of corresponding mRNAs [11].

Long-read sequencing means that a complete sequence of the transcript with all exons and all associated splicing junctions is captured within one read [12,13]. Novel long-read sequencing technologies can overcome some limitations of short-read RNA-Seq approaches in studying complex eukaryotic transcriptomes. We could also expect this approach to be very helpful in identifying positions of transposons within transcripts; repetitive nature and unpredictable insertion of transposons make their mapping in sequencing data particularly hard [14], and longer reads would make it easier.

Although Nanopore long-read sequencing has some limitations compared to short-read methods, such as the high error rate during the base-calling step, it is a user-friendly method allowing multiplex sequencing. Significant advantages of Nanopore have been successfully demonstrated in identifying novel transcript molecules and complex isoforms [13,15] or in quantifying long non-coding RNAs [16]. Moreover, thanks to its ability to perform rapid long-read sequencing analysis requiring minimal additional lab equipment or technical expertise, MinION has been used to diagnose viral diseases [17], as well as for users who would like to identify and quantify full-length transcripts, and are interested in differential gene expression. For example, MinION was used for the sequencing of human primary cardiac fibroblasts [18] and in the analysis of mechanisms of abdominal aortic aneurysm (AAA), a serious disease with a high mortality rate [19].

The primary aim of the present work was to estimate whether the number of reads obtained by sequencing using the ONT MinION platform is sufficient to characterize transcriptome changes in primary neuronal cultures (PNC) as a result of their activation by the application of PTX (an inhibitor of GABAergic synaptic transmission). If MinION provides enough reads, the secondary aim was to identify alternative splicing events happening as a result of PTX activation.

We chose the PTX activation model because a PTX-induced blockade of GABA and glycine receptors [20] causes a lack of inhibiting signals from interneurons resulting in neuronal excitation caused by excitatory postsynaptic potentials [3]. This is similar to what happens normally in the living brain, while other models of in vitro neuronal activation are less subtle. For example, commonly used KCl application simply causes depolarization of a whole neuron [21], while bath glutamine application stimulates extrasynaptic NMDA receptors rather than synaptic ones [22]. Another commonly used GABAR antagonist, bicuculline, also causes inhibitory interneuron blockade and action potential bursts in culture [22], but PTX is more stable than bicuculline [23].

We prepared 12 cDNA libraries from poly(A)+ RNA extracted from rat primary hippocampal neuron cultures. In this work, we used a specific approach to exclude uncapped mRNAs during the library preparation to enrich the samples with full-length transcripts. We then performed long-read sequencing on the MinION platform. Overall, 23,652 novel transcripts were identified. We also estimated differential gene expression at 30 min, 1 h, and 5 h time points after the PTX application by sequencing and performed PCR quantification of selected genes to confirm their differential expression.

## 2. Materials and Methods

Figure 1 summarizes the experimental design.

### 2.1. Primary Hippocampal Neuron Cultures and Their Activation with PTX

Primary neuronal cultures were prepared from the hippocampi of newborn Wistar rat pups based on previously reported protocols [24,25]. Briefly, the dissected hippocampi were trypsinised with 0.25% trypsin (MP Biomedicals, Solon, OH, USA), washed with sterile DMEM (PanEco, Moscow, Russia) containing 15 mM HEPES pH 7.0, and resuspended in the neurobasal medium (NBM; Gibco, Grand Island, NY, USA) supplemented with B27 (Thermo Fisher Scientific, Grand Island, NY, USA) and GlutaMAX (Thermo Fisher Scientific, Grand Island, NY, USA). After the cell clumps and debris were allowed to sediment, a portion of suspension was transferred to the new tube and its volume was replaced by fresh neurobasal medium (NBM) for 4–5 times. The cells were then counted and plated at a density of 250,000 cells per well on Poly-L-lysine hydrobromide 1 mg/mL (PanEco, Russia) coated 4-well plates (SPL Life Sciences, Pocheon-si, Republic of Korea). One hour after the cell plating, the medium was changed to fresh NBM. Hippocampal neurons were cultured in 1 mL of NBM for two weeks at 37 °C in a 5% CO_2_ incubator and supplemented with 1/3 of fresh medium every 3 days starting from DIV7. On DIV15, the PTX activation (the final concentration of PTX was 30 μM; PTX was dissolved in DMSO, Sigma-Aldrich, Saint-Quentin-Fallavier, France) was performed. Cells were incubated with PTX for either 30 min, 1 h, or 5 h before the cultures were harvested in QIAzol (Qiagen, Hilden, Germany) lysis reagent. Control samples had no PTX treatment.

### 2.2. Immunocytochemistry, Fluorescent Microscopy, and Image Processing

Immunocytochemical staining with antibodies to the α-subunit of CaM kinase II (CaMKIIα) was performed to identify the predominance of neuronal cells in the cellular composition and to confirm morphology stability after the PTX exposure.

On DIV15, primary neuronal cultures were fixed for 10 min with 4% neutral buffered formaldehyde solution (Biovitrum, St. Petersburg, Russia). Cells were washed twice for 5 min in 0.5 mL ice-cold phosphate-buffered saline (PBS). The cultures were treated with 0.1% PBS-Triton X100 solution and incubated for 15 min at room temperature on a shaker at 70 rpm. Each culture was washed three times for 5 min in 0.5 mL PBS. To block non-specific binding sites, cells were incubated for 1 h at RT in blocking solution (5% NGS in 0.05% PBST buffer) on a shaker at 70 rpm. An amount of 250 μL of the primary antibody solution CaMKIIα monoclonal antibody (clone 6G9, MA1-048; 1:500; Thermo Fisher Scientific, USA) in blocking solution was used for 1 h hybridization at RT on a shaker at 70 rpm. Each well was washed three times for 5 min in 0.5 mL PBST. All subsequent actions were carried out in the dark. For hybridization with secondary antibodies, each culture was covered with 250 μL of Alexa Fluor 546 Goat anti-mouse secondary antibody (Abcam 1/2000) and incubated at RT for 1 h. Each well was washed three times for 5 min in 0.5 mL PBST. PBST solution was replaced with DAPI solution (1/2000; Cell Signaling, Danvers, MA, USA) to stain the nuclei and incubated for 2 min, then washed twice for 2 min in 0.5 mL PBST. The mounting medium (Polysciences, Warrington, PA, USA) was applied to prepare slides for microscopy. Finally, slides were stored in the dark at +4 °C. Fluorescent microscopy was performed on Keyence BZ-9000E BioRevo (Keyence, Osaka, Japan) with oil immersion at ×60 microscope objective. Fiji software (ImageJ v.1.53p) was used for image processing.

### 2.3. RNA Isolation and Reverse Transcription

RNA isolation was carried out according to the improved method of RNA phenol-chloroform extraction [26,27]. All samples were treated with DNase I (Thermo Fisher Scientific, Carlsbad, CA, USA; 1 U/reaction) at 37 °C for 30 min, followed by enzyme inactivation at 70 °C for 20 min. Total RNA was then purified with Agencourt RNAClean XP beads (Beckman Coulter, Brea, CA, USA). Aliquots were taken as RT(-) controls for RT-qPCR. The RNA concentration was measured by estimating light absorption at 260 nm using NanoPhotometer^®^ N50 (IMPLEN, München, Germany). The RNA integrity number (RIN) was assessed on 2100 Bioanalyzer using Agilent RNA 6000 Pico Reagents and Chips (Agilent Technologies, Lithuania, Germany). 

In total, 150 ng of total RNA were reverse transcribed using MMLV RT Kit (Evrogen, Moscow, Russia) and random decamer primers. The resulting cDNA was diluted with three volumes of MQ water and used as a template for RT-qPCR.

### 2.4. RT-qPCR

We selected IEG *Fos* (Fos proto-oncogene) as the target gene, and *Osbp* (oxysterol binding protein) as the reference gene. Primer sequences are specified in the Table 1.

qPCR reactions were assembled using a qPCRmix-HS SYBR commercial kit (Evrogen, Russia) and incubated in CFX384 Touch Real-Time PCR System (Bio-Rad, Hercules, CA, USA). Reaction volume was 10 µL (8 µL master mix and 2 µL cDNA template). Three technical replicates were used for each sample. The following protocol for PCR was run: 95 °C for 5 min (once)95 °C for 30 s, 63 °C for 30 s, 72 °C for 30 s (41 cycle)Melt curve (65 °C to 95 °C, increment 0.5 °C for 5 s)

The results of qPCR were processed by CX Manager™ 3.1 Software (Bio-Rad, USA), and gene expression fold change was calculated in Microsoft Excel using the Pfaffl method [28]. Each experimental sample had its own corresponding control sample. Statistical analysis was performed using RStudio 1.4.11.06.

### 2.5. Full-Length Enriched cDNA Library Preparation for Oxford Nanopore Sequencing

To reverse transcribe RNA and amplify full-length cDNA, TeloPrime Full-Length cDNA Amplification kit V2 (Lexogen, Wien, Austria) was used according to the manufacturer’s protocol. This kit allows the selective reverse transcription of capped, poly(A)+ RNA molecules. For each sample, 250 ng of total RNA was taken. The concentration of the resulting cDNA was measured using a Qubit 2.0 fluorometer and Qubit dsDNA HS Assay kit (Thermo Fisher Scientific, Eugene, OR, USA). The quality of full-length cDNA was analyzed using 2100 Bioanalyzer and Agilent High Sensitivity DNA Reagents and Chips (Agilent Technologies, Lithuania, Germany). The average length of amplified cDNA was 1–3 kb.

For the preparation of barcoded cDNA libraries for Oxford Nanopore Sequencing, the Native Barcoding Expansion 1-12 (EXP-NBD104) and Ligation Sequencing Kit (SQK-LSK109) (Oxford Nanopore Technologies, Oxford, UK) were used according to the manufacturer’s protocol. Each library pool consisted of four barcoded samples (control, 30 min, 1 h, and 5 h of incubation with PTX) and corresponded to one biological replicate. The FLO-MIN106 (R9.4.1) flow cell was prepared and loaded using a commercial Ligation Sequencing Kit (SQK-LSK109) and Flow Cell Priming Kit (EXP-FLP002) (Oxford Nanopore Technologies, UK) according to the manufacturer’s protocol.

### 2.6. MinION Sequencing and Data Processing

Nanopore libraries were sequenced using a MinION Mk1B sequencing device with R9.4 flow cells. Sequencing was controlled and data were generated using ONT MinKNOW software (v3.4.12). Runs were terminated after 48 h and FAST5 files were generated.

### 2.7. Read Acquisition and Quality Control

Raw signal outputs from Oxford Nanopore Minion cells were basecalled and demultiplexed using Guppy software 5.0.16 [29]. Acquired reads were further trimmed from residual barcodes and library preparation adapter sequences using PoreChop [30] and quality-filtered using NanoFilt [31]. For NanoFilt, we used parameters *-q 10 -l 200 --headcrop 10 --maxlength 10,000*, to retain reads with a Q-score no less than 10, length no more than 10 kb, and cut low-quality starting bases.

### 2.8. Read Alignment and Counting

To align reads to the base mRatBN7 reference transcriptome (for the purpose of counting primary alignments as priority), we used minimap2 [32] with the following parameters: minimap2 -ax map-ont. To align reads to the custom transcriptome (for the purpose of estimating transcript abundance), we used minimap2 with parameters minimap2 -N 100 -p 1 -ax map-ont as recommended in [33,34]. Resulting BAM files were sorted using samtools [35] and used for read counting with Salmon ver 1.7.0 [36], with parameters suitable for long-reads: --libType A --noLengthCorrection --noErrorModel. Salmon counts were imported into the R programming environment and counted using the tximport package [37].To align reads to a reference genome (mRatBN7), we used minimap2 with the parameters *-ax splice -uf --cs --MD*. Splice junctions from the mRatBN7 GTF-file were added through the *--junc-bed* parameter. Genome-aligned reads were counted using FeatureCounts [38] with mRatBN7 annotation and parameters, allowing long-reads and read overlap: *-O --fraction -L --primary*. 

Calculations of primary-to-secondary alignment statistics and read percentages per various coverage thresholds were performed on BAM files using pysam [35,39,40] and GenomicFeatures R package [41], respectively, as well as custom scripts.

### 2.9. Transcriptome Assembly and Annotation

For custom transcriptome assembly, reads obtained from 20 cDNA libraries prepared from control and PTX-treated cultures used in similar experiments were used. For differential gene expression analysis, 12 of these libraries from the experiment with optimized PTX activation protocol were selected.

We compared four different methods of long-read transcriptome assembly: fFLAIR, relaxed and stringent mode [42], FLAMES [43], Stringtie2 [44], and Bambu [45], each with the default parameters. The effectiveness of these methods for a locus-level reconstruction of transcripts was assessed using GffCompare v0.11.7 [46] by directly comparing the raw output GTF from each assembly to Rn6 and mRatBN7 GTF files, subsetted to ~5600 most expressed genes based on the FeatureCounts output. We defined these most expressed genes as having at least five reads in each condition analyzed (Control, 30 min, 60 min, 5 h). The GTF file from the most effective method at reconstructing mRatBN7 transcripts was used to create the final comprehensive GTF file by merging it with full mRatBN7 GTF and excluding redundant entries (GffCompare with parameters -D -S --strict-match -C). Corresponding transcript sequences for this GTF, obtained by gffread [46], were used as a reference for mapping with minimap2, indexing, and read counting with Salmon. 

Protein domain annotation was performed using Conserved Domain Database (CDD) as a resource and command cddsearch, filtering results on only PFAM domains. For the repeat-family search, we used the Dfam database [47] and the program *hmmer* with parameters *--incE 0.0000000001 --dna* on all novel transcripts using a table of hits (*--tblout*) as an output. Coding potential analysis was performed using CPAT [48]. 

### 2.10. Differential Gene Expression Analysis, Visualization and Gene Ontology Analysis, and Differential Transcript Usage Analysis

Gene-level differential expression was analyzed using DESeq2 R-package [49] on transcript counts/abundances, obtained with Salmon. Adjusted *p*-values < 0.05 for the 30 min time point and <0.1 for 1 h and 5 h time points were used. (We found no DEGs with adjusted *p*-value < 0.05 at 1 h and 5 h, so we decided to use a relaxed *p*-value threshold to identify trends in gene expression at these time points).

Gene Ontology/KEGG category search was performed using clusterProfiler R-package [50], with adjusted *p*-value < 0.05. Ribosomal and endocytic KEGG pathways were visualized with the Pathview [51] webpage. Gene expression visualizations were performed using R-packages ggplot2, ComplexHeatmap [52], and VennDiagramm [53]. Regularized logarithm expression values were used for constructing gene- and family-level TE Heatmaps.

Differential transcript usage (DTU) was assessed with IsoformSwitchAnalyzer [37,54,55] on the same Salmon-generated counts.

### 2.11. Digital PCR

To verify the results obtained by Oxford Nanopore Sequencing, cDNA samples obtained using a TeloPrime kit were tested by QIAcuity digital PCR (dPCR). Expression levels of genes *Rps3a* (Ribosomal protein S3A), *Rps8* (Ribosomal protein S8), *Rack1* (Receptor for activated C kinase 1), and *Rpl15* (Ribosomal protein L15) were estimated in “control” and “30 min PTX” samples. *Hprt* (hypoxanthine phosphoribosyltransferase 1) was chosen as the reference gene because its expression showed the least difference between groups in our sequencing data from this and other experiments performed in the laboratory. Additionally, probes were designed for the multiplex analysis. The following fluorophores were used for probes: *Rps3a*—ROX, *Rps8*—Cy5, *Rack1*—TAMRA, *Rpl15*—ROX, *Hprt*—FAM.

Primer pairs and probes are shown in Table 2.

The genes listed above are highly expressed. Since the dPCR lower detection limit is 1 target copy per 1 μL of the reaction mixture, optimal dilutions for these abundant targets were selected based on the preliminary experiments. We aimed to register no more than 25% positive events in each sample. 

RT(-) and “no template” control samples were processed in the same plate with experimental samples. The total pre-mix volume was 12 μL (3 μL 4 × concentrated Master Mix (Qiagen, Germany), 2 μL template DNA, forward and reverse primers to the final concentration 0,8 µM, probes to the final concentration 0,4 µM, 5,07 μL nuclease-free water). Sample mixes were transferred to the Nanoplate 8.5K 24-well (Qiagen, Germany) and coated with a rubber seal (Qiagen, Germany) using the roller. Each sample had three technical replicates. The nanoplate was gently transferred to the QIAcuity One 5plex thermal cycler amplifier (Qiagen, Germany), and the following protocol for PCR was run: 95 °C for 2 min—PCR initial heat activation; once95 °C for 30 s, 60 °C for 30 s—amplification for 45 cycles40 °C for 5 min—cooling; once

Imaging was performed with the following parameters: Exposure duration/Gain: green channel—500 ms/6, yellow channel—500 ms/6, orange channel—400 ms/12, red channel—300 ms/8, crimson channel—400 ms/16.

Raw dPCR data (template DNA copy number per μL) were obtained with QIAcuity Software Suite v. 1.2 (Qiagen, Germany). Thresholds were automatically identified by the software for each fluorescent channel. Proportional change in genes expression for “Control” and “30 min PTX” groups normalized to the control group value was then calculated in Microsoft Excel. The number of cDNA copies for each gene of interest was normalized to the average number of *Hprt* cDNA copies in the same sample. 

## 3. Results

### 3.1. Fos Expression Level Increases after the PTX Application

To verify the PTX effect on PNC and standardize the samples before the preparation of cDNA libraries for sequencing, we measured the expression of *Fos*, a well-known immediate early gene associated with neuronal plasticity [56]. *Fos* expression level in activated cultures was evaluated by RT-qPCR. Figure 2 shows *Fos* expression dynamics in primary neuronal cultures activated with PTX. Incubation with PTX causes a sharp increase in *Fos* expression apparent by the 30th minute; increased *Fos* expression is maintained for at least an hour after the PTX application but shows some decrease at the 5 h time point.

### 3.2. Custom Transcriptiome Assembly Based on the Sequencing Data

In total, we sequenced 20 cDNA libraries (from PTX-treated and untreated cultures), prepared with Lexogen TeloPrime v2 Kit. Average length of reads per sample ranged from 1000 to 2600 nt., with the average Phred score being approximately 20 (Appendix A). To further check the length of acquired cDNA in comparison to reference transcripts, we aligned reads to the mRatBN7 transcriptome assembly (Figure 3a,b) and then calculated the “primary” alignment (either unique, or best alignments across multiple hits) statistics of their sequence similarity to the reference (Figure 3a). In total, 99% of reads in each sample had at least one primary alignment assigned (Appendix A). After averaging the data across all samples, we found that only 63% of primary alignments were covering more or equal to 80% length of corresponding reference transcripts (Figure 3b). Analysis of reference coverages at different similarity thresholds showed that transcripts coverage lowered linearly down to 46% of all reads at 0.9 (full coverage by read), thus suggesting that only half of all reads could possibly be full-length. Read-to-transcript similarity at 0.9 (full “inclusion” of read in reference) also dropped to only 70% of all reads (Figure 3c). This suggests that, despite being only a part of a reference sequence, the rest of them (10–30%) could contain additional information from the same or similar loci, not yet present in the current base transcriptome.

To examine this possibility, we used reads from all 20 samples to prepare several different transcriptome assemblies using four distinct long-read assembly pipelines (Figure 3d): FLAIR, relaxed, and stringent mode [42], FLAMES [43], StringTie2 [44], and Bambu [45]. To choose the best performing assembly pipeline, we compared transcript models from each one to reference models of transcripts from the ~6k most abundantly expressed genes, using the locus-level statistic of GffCompare [46]. As our experiment was spread over a long period of time, we performed this step twice: first with Rn6 (Appendix A) and later with mRatBN7 genome (Figure 3d) versions and for our final mRatBN7-based transcriptome compared only FLAIR stringent, FLAMES, and Stringtie2 methods (see the Section 2). We did not recreate our analysis with the Bambu transcriptome on the mRatBN7 genome basis, because at the time our experiment was performed, the Bambu pipeline lacked a standalone GTF output (not merged with reference) and thus was not suited for this comparison. For the rest of the analysis, we chose the FLAIR stringent pipeline, due to its second best score in sensitivity, comparable only to the same model with relaxed criteria (with Rn6 genome) and the best one with mRatBN7, while maintaining a decent precision level. As an additional consideration, we have taken into account the detection of novel intergenic transcripts/loci, which were consistently assembled by three different methods (FLAIR, Stringtie, Bambu), but not detected by FLAMES (Appendix A). 

We combined the resulting assembly with mRatBN7 base transcriptome and excluded redundant transcripts, which in the end gave us 73,728 transcripts in total, out of which 23,652 were novel. At the next step, we classified these transcripts using GffCompare software v0.11.7 [46], which assigns them distinct codes based on the difference in splicing when compared to known transcripts (Figure 4a). The majority of assembled transcripts were different from the reference only in splice junctions (~10,000; code “j”); less abundant groups were novel isoforms containing new retained introns (codes “m”, “n”), transcripts with the same structure but end length difference (“=−”,”=+”), and completely novel loci (2000–3000 each, Figure 4a). Protein-size ORFs were detected in most of the new isoforms, suggesting that these transcripts likely encode protein sequences (Figure 4b), with the notable exception of “novel loci” that either included only small ORFs or were noncoding. Protein domain frequency analysis (PFAM) [57,58] in each transcript group (Figure 4c,d) showed the unexpected abundance of the RNA-binding domain (RRM_1) hits among new transcripts; RRM_1 was among the top five most frequent unique (one per transcript) domains per group. To further examine this possibility, we performed hypergeometric testing on domain counts and found significant overrepresentation of RRM_1 among all five novel groups (Benjamini–Hochberg *p*.adjusted < 0.05), while no significance was shown for baseline mRatBN7 (“Full match”) transcripts.

### 3.3. The Majority of Novel Loci Transcripts Are Transposon-Related

Despite the low coding probability of most of the novel loci, some of them also harbored known protein domains (Figure 4d). Notably, we observed among them MLVIN_C (murine leukemia virus integrase C-terminal domain) and TLV-coat (viral surface polyprotein) domains, suggesting their transposon or endogenous retrovirus heritage. 

To examine this possibility, we analyzed the presence of repeat-related sequences from the DFAM database across the transcripts of all groups. We observed a large predominance of repeat-related sequences in the “novel loci” group (Figure 5a), while a much lower number was found for other groups, with the only exception being the “novel junctions” group due to its initially high number of transcripts. The percent of repeat-containing transcripts (Figure 5b, shown in red and blue) was noticeably higher for all subgroups of the “novel loci” group, with the “novel junctions” subgroup of the “novel isoforms” group having the lowest number of repeats. Interestingly, the vast majority of all repeat hits per transcript mapped to 3’-UTR regions for each group with significant coding probability. The “novel loci” group had no such noticeable distinction (Figure 5c). This may be explained by most of the “novel loci” transcripts having larger numbers of very short nonproductive ORFs.

### 3.4. 30-Minute PTX Incubation Leads to Massive Ribosomal Gene Downregulation in Neuronal Cultures

Using our custom new assembly as a reference, we performed differential expression analysis of samples chemically activated by PTX and fixed at different time points (30 min, 60 min, 5 h after the activation; three samples per each time point including the control; 12 libraries in total). Unexpectedly, we found that a large total number of DEGs could be detected as early as 30 min (1278 genes upregulated, 800 downregulated, *p*.adj < 0.05), but that number abruptly dropped to 446 at 60 min; at 5 h, only 13 DEGs were identified (Figure 6a,b). Examination of known early response genes (IEGs) to gauge the extent of neuronal activation (Figure 6c, Appendix A) also showed their peak expression at 30 min, which is concordant with the previous data [59,60], with the only exception being *Gadd45* family (Figure 6c). Despite the very low number of late response genes, we also observed some very well-known participants in neuronal activity at later stages, like *Bdnf* [61].

To profile the functions of numerous genes differentially expressed at 30 min after the PTX application, we performed Gene Ontology [62,63] (Figure 7a) and KEGG-pathway [64] (Appendix A) analyses. Surprisingly, we found that top GO categories for both up- and downregulated genes were directly related to protein metabolism. More specifically, genes associated with protein transport and localization were upregulated while ribosomal ones were downregulated (Figure 7a). In the case of ribosomal proteins, expression change was evident for more than half of all existing genes in the pathway (Figure 7b,c), even if their mean fold change was noticeably lower (~1–2) than for the upregulated group (Figure 7a). We also noticed downregulation of genes related to ATP metabolism, mostly represented by mitochondrial genes highly expressed at baseline (Figure 7c). Endocytosis pathway genes underwent massive expression level changes as well, with most genes in the pathway being upregulated (Appendix A; Figure 7c).

For a few selected genes encoding ribosome and ribosome-associated proteins, their downregulation in neuronal cultures exposed to PTX for 30 min was confirmed with dPCR (Appendix A).

### 3.5. 30-Minute PTX Incubation Does Not Cause Notable Increase in Alternative Splicing

Finally, we attempted to analyze differential transcript usage (DTU) as a result of PTX treatment (Appendix A). Unexpectedly, we only detected minor changes at all time points (16, 16, and 4 affected isoforms in 30 min, 60 min, and 5 h, respectively (Figure 8a)). However, there was a noticeable transcript switching for *Gabrb1* gene (Figure 8c). This gene encodes GABAR subunit β1, and GABAR is a direct target of PTX.

## 4. Discussion

In this study, we performed long cDNA library preparation and Oxford Nanopore sequencing of RNA extracted from rat neuronal cultures to expand the knowledge of rat neuronal transcriptome and its changes in response to neuronal activity. We identified a relatively high number of transcripts that were not reported previously. We found that most of these transcripts either showed only a minor difference from the currently used mRatBN7 transcriptome assembly (primarily in splice junctions), or belonged to entirely novel, transposon-derived, and largely non-coding loci. Such results could indicate that long-read sequencing is still able to provide new information about the variable length of known exons or whole transcripts. In our case, the results could be biased towards neuron-specific isoforms since we performed our experiments exclusively on neuronal cultures. Nevertheless, we should note that an abundance of predominantly small-length (novel junctions) transcript variations could also be partly caused by known Oxford Nanopore high sequencing error rate [66].

Concerning transposon transcripts, neurons are known to abundantly express various mobile elements, including L1, in non-pathological conditions [6], and in our experiment L1 sequences were particularly enriched in the “novel loci” group compared to all other groups (Figure 5a). L1 mRNA is also known to be polyadenylated [67], similarly to some long non-coding RNAs [68], which could also explain the observed abundance of various non-coding and/or transposon sequences even after the poly(A)+ enrichment.

To examine the potential of the Oxford Nanopore sequencing method to detect gene expression changes associated with neuronal activity, we applied this method to examine neuronal cultures at different time points after PTX application, and, surprisingly, observed massive differential gene expression as early as 30 min after the activation. This transcriptional shift coincided with peak expression of classical neuronal activity markers—IEGs [59,60], and, according to Gene Ontology analysis, mainly involved genes regulating protein metabolism. This early transcription upregulation already showed signs of returning to baseline at the 1 h time point. To our knowledge, observation of such extensive expression change at such an early time is uncommon, but it does agree with the need of protein synthesis and degradation in memory formation and long-term potentiation processes [69,70]. In our case, genes involved in the regulation of protein membrane transport/endocytosis were among the top enriched upregulated categories, which could possibly indicate that their early activation is crucial to prepare neuronal cells for timely receptor profile switching at later time points, when receptor proteins are actually synthesized. However, we did not observe any comparable delayed transcriptional response at 5 h reported by other groups [71], except for a few known late-response genes like *BDNF* [61], which could be possibly explained by the receptor-specific profile of such a second wave of activation, possibly involving NMDAR activity. 

Concerning downregulated categories, we observed decreased expression of various ribosomal subunit genes at 30 min after the PTX application, even though the degree of change for individual genes from this group was relatively small. Despite such minor changes, ribosomal protein expression shifts may also be connected to activity-induced proteometabolic changes—specifically, translational specification strategy by optimizing ribosomal composition. Ribosomal protein specification was already shown for axonal and dendritic compartments, where these proteins undergo local translation [11] and could participate in the preferential translation of synaptic mRNA subset [72]. We may also speculate that since ribosomal protein synthesis may be increased on the translational level to repair damaged ribosomes in dendrites [11], temporarily decreased transcription of the corresponding mRNAs is likely not detrimental for neuronal ribosomes and may even be used to redirect transcriptional machinery to activity-induced genes. Notably, RPS13 gene expression was not affected in our experiment; this gene was previously reported to have stable expression in various normal and tumor tissues [73].

While ribosomal genes present a distinctive group among all the genes, we found to be downregulated at the 30 min time point, many other “housekeeping” genes, like genes involved in ATP metabolism (Figure 7c) were also downregulated. We suggested that this downregulation may reflect a shift of transcriptional resources towards activity-induced genes without actually reducing concentrations of proteins encoded by the temporarily downregulated housekeeping genes. Indeed, the majority of downregulated genes in our case had much higher baseline expression than the upregulated group (Appendix A). Both ribosome- and mitochondria-related genes are known to be very abundantly expressed throughout the cell’s life, which leaves only two possible options for their downregulation mechanics: either by induced mRNA degradation of presynthesized transcripts, or by arrest of their ongoing, constitutive transcription. Given the latter possibility, we could hypothesize that activity-induced neuronal gene expression may require partial diversion of transcriptional machinery from constitutive to newly upregulated genes, resulting in mild transcriptional slowdown of the former. If this is true, the availability of common proteins, like general transcriptional factors, nucleosome motors, or clusters of Pol II [74], could possibly be a limiting factor in activity-dependent conditions, resulting in global transcriptional dampening in favor of current relevant hotspots.

Unexpectedly, we found that alternative splicing as a result of PTX activation is in fact a rare occurrence. We found no pattern in the few mRNAs that underwent alternative splicing in our experiment (listed in Appendix A). The most noticeable result here was a transcript switching for *Gabrb1*, a GABA signaling-related gene.

We conclude that MinION sequencing has proven to be an adequate method for assessing differential gene expression in primary neuronal cultures after their activation. We suggest, however, that combining this method with high-throughput short-read sequencing would provide even more accurate results.

### 4.1. Limitations of the Study

We should note that such extensive early activity-dependent changes as those reported in our work are uncommon. To our knowledge, many other groups did not observe such changes in similar experiments. One recent study [71], which examined the time course of gene expression in response to neuronal activation (0–6 h after KCl application), reported only 19 differentially expressed genes before 1 h, most of these being IEGs (termed “rapid primary response genes”, rPRGs). Other studies, examining more remote time points, reported varying numbers: 47 genes 1 h after the application of bicuculline [75], 877 genes (in either one or both models) 2 h after the application of glutamate or bicuculline [76], 931 genes after 2 h incubation with PTX/forskolin/rolipram [77], and almost 2000 genes at 1 h or 6 h time points after KCl application [78] (Appendix A). Upregulated genes identified in our study are more numerous than usually reported, and were detected much earlier in time compared with data from other studies. This is not easy to explain, given that RNA polymerase II kinetics suggests that this enzyme may not be fast enough to perform complete transcription of the longest genes in our list (400 kb) in this time. One possible explanation of this could be related to spontaneous neuronal activity, which was not blocked in our experiment and could make such genes more prone to transcriptional bursts due to residual pre-associated transcriptional complexes. The other explanation could be possibly related to our choice of full-length Oxford Nanopore sequencing methodology, which is more sensitive to small changes in mRNA content due to less ambiguous sequence alignment. 

Genes identified as differentially expressed in our study are not very consistent with DEGs reported by other researchers. However, it should be pointed out that DEG lists in similar experiments generally show weak consistency between studies (Appendix A), possibly due to the fact that most of these studies were performed on very different model systems, including human iPSC-derived neurons, neurons from the cortex and hippocampus of rats, and murine neurons of various DIV (from 6 to 20). Additionally, different approaches to neuronal activation used in these studies utilize different molecular mechanisms. Elevated extracellular KCl causes depolarization of the whole neuron (but often does not trigger neuronal activation) and induces CaMKII, CaMKIV, ERK, and calcineurin signaling pathways [21]. It is well-known that NMDA receptor activation induces LTP via calpain, PKA, CamKII, ERK, and some other pathways; these data were obtained on different experimental models [79]. However, molecular mechanisms in neuronal culture may not be the same as processes observed in brain structures. A study dedicated to glutamate receptors in cultured hippocampal neurons showed that while synaptic NMDA receptors (activated in culture by bicuculline-induced GABAR blockade and bursts of action potentials) promote nuclear signaling to CREB, induce BDNF gene expression, and activate an anti-apoptotic pathway, bath glutamate application causes stimulation of extrasynaptic NMDA receptors which trigger exactly the opposite effects [22].

Notably, expression changes of ribosomal protein genes were also seen in two out of three other studies where PTX was applied (for 2, 15, and 48 h) [77,80,81]; however, in contrast to our results, major upregulation of these genes was observed at different time points (2 h and 15 h). We verified some of our results by dPCR, which excludes long-read sequencing bias; still, we could not exclude the possibility of these results reflecting some processes that are specific only to earlier PTX activation time points, absence of spontaneous activity blockade, or the protocol of culture preparation used in our study.

Another possible concern is that while we aimed to only include full-length transcripts when preparing cDNA libraries, the median length of our reads was about 1.2 kb which is considerably shorter than the average rat mRNA length, 2.32 kb [82]. It is possible that during the cDNA amplification step in the library preparation, shorter fragments had an advantage [83], so the ratio of different transcripts in our results was biased towards shorter fragments. Short-read sequencing suggests nucleic acid fragmentation prior to cDNA amplification, so there would be no such bias. However, we took the necessary precautions to avoid overcycling and PCR product length disproportion while preparing libraries by using qPCR to determine the optimal endpoint PCR cycle number before the whole library amplification, according to the manufacturer’s protocol. It also must be noted that other studies where transcriptome sequencing with MinION was performed report a similar or shorter average read length [13,15,18,19] and that some of these authors claim that these results represent a full-length transcriptome [13,19].

### 4.2. Possible Glycine Receptors Target Side Effects and Perspectives for Future Research

While being a major neurotransmitter in the brain stem, glycine performs diverse roles in the forebrain and hippocampus as well, including tonic inhibition via extrasynaptic glycine receptors, NMDAR-dependent plasticity as a coagonist, and GlyR-mediated co-inhibition of GABAergic inhibition [84]. Glycine receptors (GlyR) are formed by the assembling of five subunits. Multiple types of GlyR subunits are known (four main types of α-subunit, with alternative splicing generating some additional variants, and one type of β-subunit). α-subunits can form homo- or heteromeric receptors in combination with the β-subunit. The β-subunit is the only one able to interact with the scaffold protein gephyrin and, hence, provides an anchoring mechanism necessary for synaptic localization and function. Receptors containing only α-subunits have been found in extra-synaptic locations [85].

Receptor composition can also vary depending on the neuronal cell type and maturation stage, exhibiting a shift from α2 homomeric receptors in embryonic retinal/brain stem neurons to α1β heteromeric in adulthood [86], and from heteromeric α1β to extrasynaptic α2/α3 composed receptors in rat adult hippocampal cells [87]. However, these results partly conflict with publicly available transcriptomic data [88], showing predominant expression of β-subunits over α-ones in mouse hippocampal excitatory and inhibitory neurons.

Although the activation pattern of PTX is quite precise in that it causes action po-tential bursts rather than depolarization of the entire neuron, some works have shown that PTX is nonselective for GABAR because it also blocks α2 homomeric glycine receptors [20]. α2 glycine receptor subunits are mostly expressed by inhibitory neurons [89] and α3 subunits are mostly expressed by excitary neurons [90]. Notably, the same concern is relevant for the other popular model of neuronal activation, application of bicuculline—the substance which was also shown to be a potent antagonist of the glycine receptors, both homomeric GlyRα and heteromeric GlyRαβ. Moreover, inclusion of β-subunits in the receptor increases the inhibitory potency of bicuculline [91]. On the other side, inclusion of β-subunit in GlyR was previously shown to confer insensitivity towards picrotoxin. Picrotoxin is ineffective as a channel blocker for GlyRαβ heteromers at concentrations ≤100 μM [92,93].

According to the current transcriptomic and single-cell NGS data about various types of neurons [94], heteromeric GlyRαβ receptors are prevalent in the hippocampus. Taken together with the fact that PTX is a more potent antagonist of GABAR than of GlyR [91], this could suggest that PTX side activity on GlyR in our cultures is quite low. Nevertheless, further research, specifically addressing such possibility in both PTX and bicuculine models and better distinguishing GlyR from GABAR effects (with the usage of specific antagonists), would be relevant in the future.

## Figures and Tables

**Figure 1 cells-13-00383-f001:**
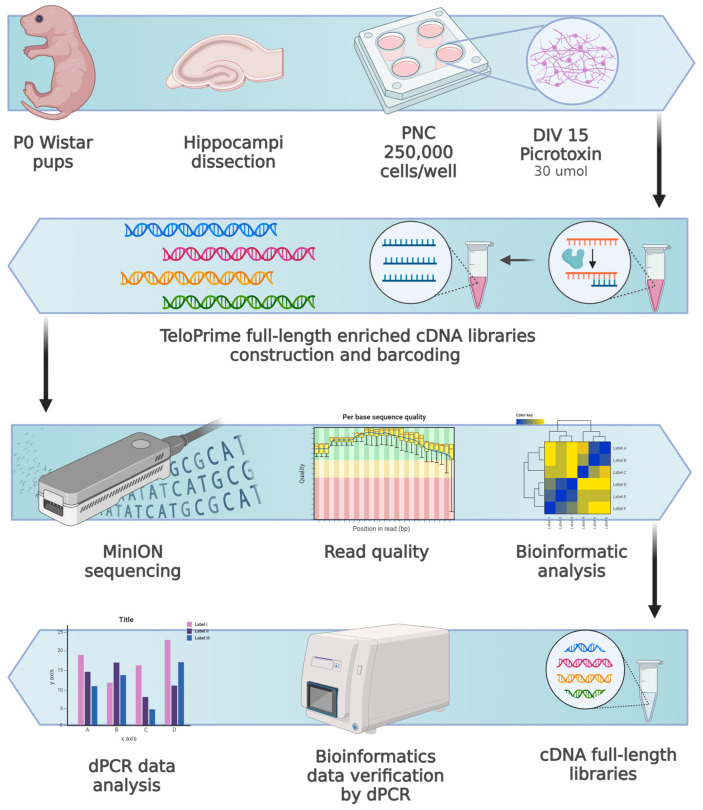
The design of the experiment. Primary neuronal cultures (PNC) were generated from P0 Wistar pup hippocampi. At DIV15, PNC were incubated with PTX (final concentration 30 umol) for 30 min, 1 h, or 5 h, followed immediately by RNA extraction from samples. From this RNA, full-length enriched (poly(A)+) cDNA libraries were constructed and barcoded, followed by the MinION sequencing. The results of the sequencing data analysis were verified by digital PCR (dPCR).

**Figure 2 cells-13-00383-f002:**
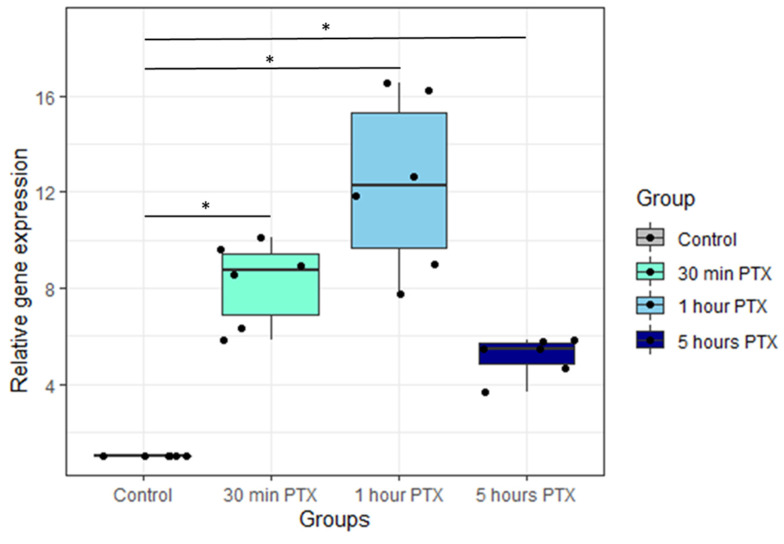
*Fos* expression fold change in PNCs treated with PTX. Fold change compared with untreated control was calculated using the Pfaffl method [28]. Boxes represent median and quartiles; whiskers represent 95% confidence intervals. *n* = 6 in each group. *—*p* < 0.05, Wilcoxon matched pairs test.

**Figure 3 cells-13-00383-f003:**
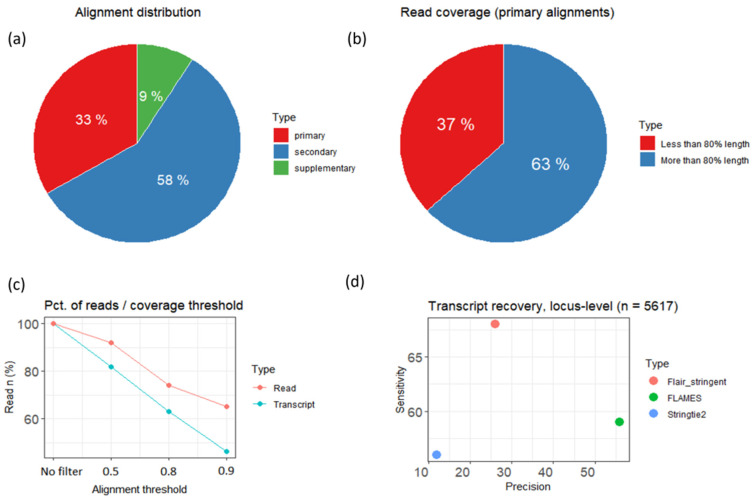
(**a**) Base alignment statistics, given as percentage of different read alignment categories, averaged across all samples (primary—unique alignment). (**b**) Percentage of primary alignments covering more than 80% length of reference transcripts. (**c**) Red line—read-to-transcript similarity at different thresholds (aligned length/total length, 0.9—read almost fully contained in reference), blue line—percentage taken by a reference sequence within each read (0.9—almost fully included in read). (**d**) Comparison of different long-read transcriptome assembly pipelines by their ability of restoring sequence of 5617 most abundantly expressed transcripts using the mRatBN7 genome assembly.

**Figure 4 cells-13-00383-f004:**
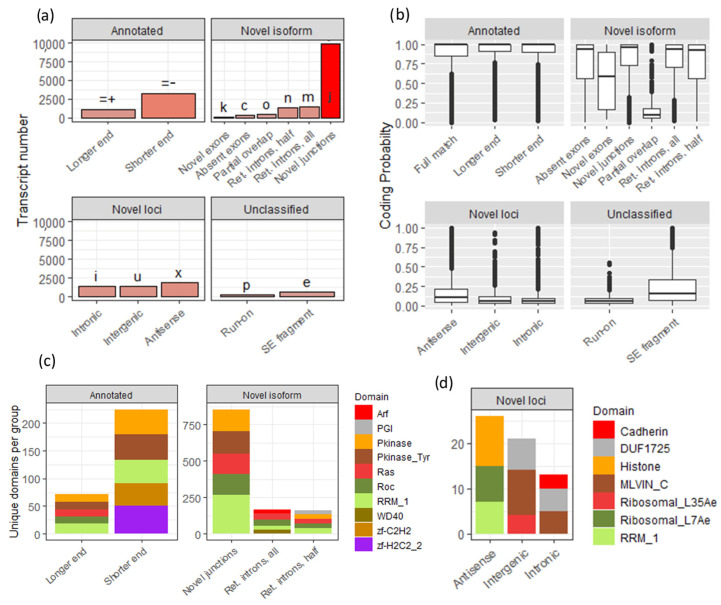
(**a**) Categorization of transcripts based on the *GffCompare* [46]. Two new groups (‘=−’, ‘=+’) were added to represent transcripts, categorized as equal (‘=’) based on their exon–intron structure, but still differing in the end section. Codes meaning: ‘k’ novel exons, ‘c’ absent exon, ‘o’ partial overlap, ‘n’ retention introns, half, ‘m’ retention introns, all, ‘j’ novel junctions (**b**) Coding probability of transcripts, averaged across groups. Codes meaning: ‘i’ intronic, ‘u’ intergenic, ‘x’ antisense, ‘p’ run-on, ‘e’ SE segment (**c**) Total number of unique PFAM domains (across top five within each subgroup) in “annotated” and “novel isoforms” groups. (**d**) Total number of unique PFAM domains (across top three within each subgroup) in the “novel loci” group.

**Figure 5 cells-13-00383-f005:**
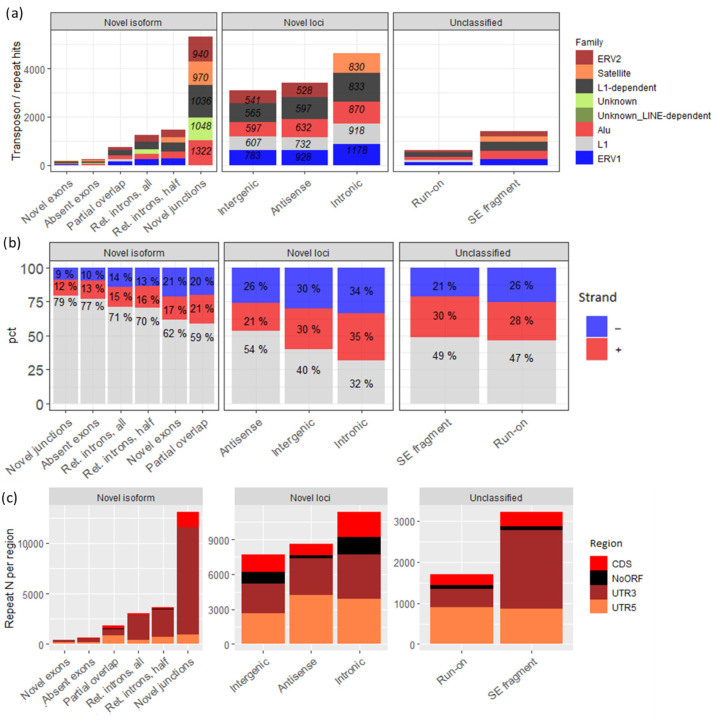
(**a**) Total, nonfiltered number of repeat hits found, averaged per group per repeat family. Numbers are shown for the four most abundant groups. (**b**) Percent (pct) of repeat-containing transcripts per group. Portions of transcripts with sense-strand hits are shown in red, antisense ones—in blue. Grey color means the rest of transcripts without repeats. (**c**) Total number of repeat hits per region (for all possible ORFs combined), shown for all groups.

**Figure 6 cells-13-00383-f006:**
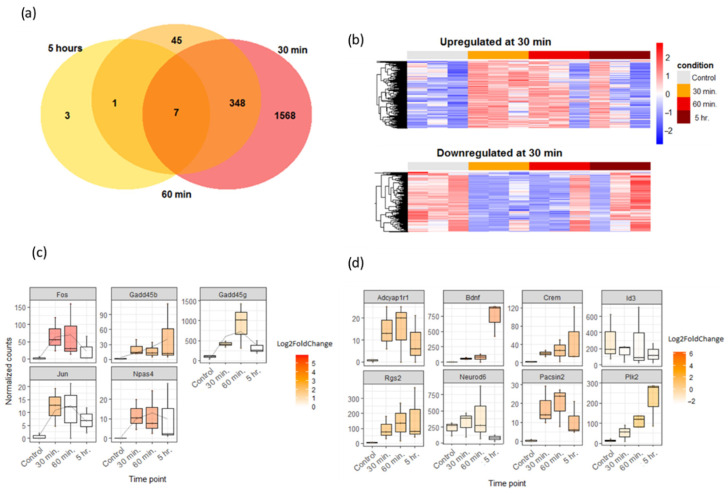
(**a**) Venn diagram of all genes differentially expressed at different time points after the PTX application. (**b**) Heatmaps showing expression levels of genes differentially expressed at 30 min across all time points. Columns represent different biological replicates. (**c**) Expression dynamics of selected IEGs across time points, AM ± SD (*p*.adj < 0.05 for 30 min, *p*.adj < 0.1 for 60 min and 5 h). Colored boxes represent significant changes (white—no significance), with hue representing the degree of difference from control. (**d**) Expression dynamics of selected late response genes, AM ± SD (*p*.adj < 0.05 for 30 min, *p*.adj < 0.1 for 60 min and 5 h). Colored boxes represent significant changes (white—no significance), with hue representing the degree of difference from control.

**Figure 7 cells-13-00383-f007:**
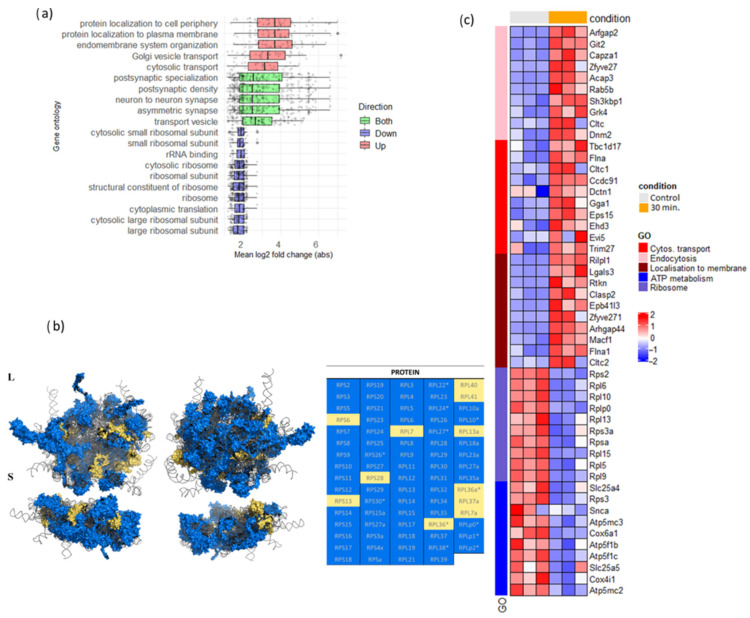
(**a**) Top 10 down- and upregulated Gene Ontology categories for DEGs identified at 30 min after the PTX application. Y axis represents GO category, X axis—log2 fold change, averaged across all genes in the corresponding category. (**b**) Visualization of the ribosomal pathway genes that are differentially expressed at 30 min after the PTX application. Structure of the mammalian ribosome, two views each for the large (L) and small (S) subunits [65]. Grey color represents rRNA. Protein products of downregulated genes are shown in blue, protein products of non-DE genes are shown in yellow. In the table, ribosomal proteins previously reported in [11] to be rapidly replaced within assembled ribosomes in neurons under oxidative stress conditions are labeled with asterisk. (**c**) Heatmaps showing top 10 DEG (sorted by log 2 fold change) per GO category.

**Figure 8 cells-13-00383-f008:**
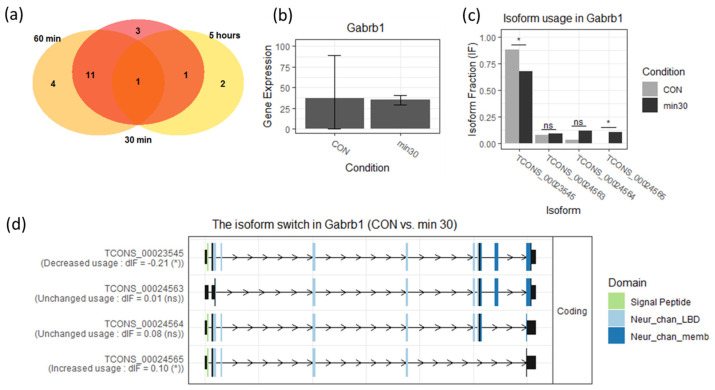
(**a**) Venn diagram for alternative splice isoforms detected at different time points after the PTX application. (**b**) Differential expression of the *Gabrb1* gene. (**c**) Differential transcript usage for the *Gabrb1* gene; ns— not significant, *—adjusted *p*-value < 0.05. (**d**) Exons present in four different Gabrb1 mRNA isoforms identified in our experiment. CON—control, min30—30 min after the PTX application.

**Table 1 cells-13-00383-t001:** Primer pairs for RT-qPCR.

Gene	Primer Pair Sequences
*Osbp*	F: 5′-TCC GGG AGA CTT TAC CTT CAC TT-3′ R: 5′-GTG TCA CCC TCT TAT CAA CCA CC-3′
*Fos*	F: 5′-CAA AGT AGA GCA GCT ATC TCC-3′ R: 5′-CTC GTC TTC AAG TTG ATC TGT-3′

**Table 2 cells-13-00383-t002:** Primer pairs for dPCR.

Gene	Primer Pair Sequences
*Rack1*	F: 5′-ATG ACC GAG CAA ATG ACC CT-3′ R: 5′-TCT CGT GGT AGT GCC CGT TG-3′ Probe: 5′-CCC GAA CAG CAG CAA CCC GCT TAT CAT-3′
*Rps8*	F: 5′-AAC CCT ACC ACA AGA AGC GG-3′ R: 5′-TAT TGC CTC CTC GAA CTC GG-3′ Probe: 5′-AGA TGG CTA TGT GCT CGA AGG CAA-3′
*Rps3a*	F: 5′-CCG ATG GGT ATT TGC TCC GA-3′ R: 5′-CCA ATG CTG TCT GGA ATC AGT T-3′ Probe: 5′-ATC CTA TGC GCA GCA CCA GCA-3′
*Rpl15*	F: 5′-TTC AGT CTG TTG CTG AGG AGA G-3′ R: 5′-TGT CGT TGT GGA CTG GTT TG-3′ Probe: 5′-AGT CCT GAA TTC CTA CTG GGT TGG TGA AG-3′
*Hprt*	F: 5′-CGT CGT GAT TAG TGA TGA TGA AC-3′ R: 5′-CAA GTC TTT CAG TCC TGT CCA TAA-3′ Probe: 5′-CCT GGT TCA TCA TCA CTA ATC ACG ACG C-3′

## Data Availability

The data that support this study are provided in the main text or the Appendix A. All raw sequencing data were deposited to the GEO resource under the accession number GSE218694.

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
