# Peer review of "Downregulation of Ribosomal Protein Genes Is Revealed in a Model of Rat Hippocampal Neuronal Culture Activation with GABA(A)R/GlyRa2 Antagonist Picrotoxin"

_cells, 2024, doi:10.3390/cells13050383_

Round 1
Reviewer 1 Report
Comments and Suggestions for Authors
In the present manuscript, Beletskiyng et al. use the Oxford Nanopore Technologies MinION-based long-16 read sequencing and transcriptome assembly to perform transcriptional 13 profiling of rat hippocampal primary neuron cultures after stimulation with picrotoxin (PTX). They used this tool to understand molecular mechanisms of neuronal activation further. Overall, they found 23652 novel transcripts, out of which ~6000 were entirely novel and mostly transposon-derived loci. Overall, they report a down-regulation of 24 genes after a 30-minute incubation with PTX. They discuss it as a possible redistribution of transcriptional resources towards activity-induced genes. They also suggest the use of the MinION platform to study neuronal plasticity.
The manuscript is generally well-written, and the author's intentions are well-explained. I concur with the use of c-fos activation as a marker of synaptic activity. However, I found the title, and in part, the conclusions, misleading since picrotoxin is a GABAa and glycin antagonist.
Although they explained why they've preferred picrotoxin over bicuculline, they cannot just consider in their results and discussion the GABAa receptor, but they must consider the glycine receptors as well.
Hence they should reconsider their results and the discussion of their results.
On a minor note Supp. Fig 6 needs a neuronal marker or improved bright field images.
The sequences of c-fos and Osbp primers are inverted.
Comments on the Quality of English Language
Th equality of English is good
Author Response
Reviewer 1
(Major concerns)
Point 1. However, I found the title, and in part, the conclusions, misleading since picrotoxin is a GABAa and glycin antagonist. Although they explained why they've preferred picrotoxin over bicuculline, they cannot just consider in their results and discussion the GABAa receptor, but they must consider the glycine receptors as well. Hence they should reconsider their results and the discussion of their results.
We thank the reviewer for the positive assessment of our work and the very helpful suggestions to improve the manuscript. We changed the title accordingly to the reviewer note and added to the discussion section our vision to this point in the revised version. Сoncerning glycine receptor/GABA receptor cross-reactivity, we have to note that we already mentioned this possiblity in our discussion section, particularly its antagonistic activity towards homomeric a2 receptors. We recently found out that, according to Allen Brain Atlas data (celltypes.brainmap.org), expression of glycine receptor b subunit (glrb gene) is much more predominant in mouse hippocampal cells compared to a2 subunit, while its incorporation into receptor complex is shown to confer insensitivity towards picrotoxin. However, such data comes in conflict with more earlier reports claiming predominant expression of a2/a3-subunit composed extrasynaptic receptors in adult rat hippocampus (Lynch J., 2009). Given the possibility of heteromeric glycine receptor compostion and higher prevalence of GABA receptors in hippocampus overall, it still could be assumed that GlyR side activity of PTX may be quite low, but nevertheless, we admit that we cannot readily distinguish individual effects of two receptors, as it requires further research. We expanded the discussion section accordingly and in our future work are planning to explore these possibilities with selective antagonists for both receptors.
Point 2. On a minor note Supp. Fig 6 needs a neuronal marker or improved bright field images.
We added to the section Materials and methods a point 4.2 Immunocytochemistry, fluorescent microscopy and image processing with description protocol for immunodetection neuronal marker CamKIIa in our neuronal cultures and added to figure S6 (C) control culture at DIV15 stained with antibodies to alpha subunit of CaM kinase II.
Point 3. The sequences of c-fos and Osbp primers are inverted.
The table has been corrected accordingly in the revised version.
Reviewer 2 Report
Comments and Suggestions for Authors
This study utilizes long-read sequencing technology to analyze gene expression in rat hippocampal neuron cultures stimulated with picrotoxin. It identifies 23,652 novel transcripts, including about 6,000 entirely new ones, primarily from transposon-derived loci. Differentially expressed gene analysis showed significant changes in gene expression shortly after stimulation, with 3,046 genes affected, notably including downregulation of genes encoding ribosomal proteins. These findings suggest a reallocation of transcriptional resources in response to neuronal activity and provide new insights into mRNA dynamics in neuronal plasticity. This approach, combined with other sequencing methods, could advance the understanding and study of neuronal plasticity.
The paper is well-written and the evidence is laid out clearly. It will be of great interest to the readers of Cells.
Author Response
We thank the reviewer for his very positive assessment of our work.
Round 2
Reviewer 1 Report
Comments and Suggestions for Authors
The authors have addressed all the issues raised in my first revision